# Chemical Evaluation of *Liquidambar styraciflua* L. Fruits Extracts and Their Potential as Anticancer Drugs

**DOI:** 10.3390/molecules28010360

**Published:** 2023-01-01

**Authors:** Rafaela G. Pozzobon, Renata Rutckeviski, Juliane Carlotto, Vanessa S. Schneider, Lucimara M. C. Cordeiro, Graziele Francine Franco Mancarz, Lauro M. de Souza, Rosiane Guetter Mello, Fhernanda Ribeiro Smiderle

**Affiliations:** 1Instituto de Pesquisa Pelé Pequeno Príncipe, Curitiba 80240-020, PR, Brazil; 2Faculdades Pequeno Príncipe, Curitiba 80230-020, PR, Brazil; 3Departamento de Bioquímica e Biologia Molecular, Universidade Federal do Paraná, Curitiba 81531-980, PR, Brazil

**Keywords:** *Liquidambar styraciflua*, antioxidant activity, antitumoral activity

## Abstract

*Liquidambar styraciflua* L. is an aromatic species, popularly used in traditional Chinese medicine to treat diarrhea, dysentery, coughs, and skin sores. The present study was designed to investigate the chemical composition and biological potential of extracts obtained from the fruits of this plant. For the chemical evaluation, it was used mainly liquid and gas chromatography, plus NMR, and colorimetric methods. The aqueous extract (EA) originated two other fractions: an aqueous (P-EA) and an ethanolic (S-EA). The three extracts were composed of proteins, phenolic compounds, and carbohydrates in different proportions. The analyses showed that the polysaccharide extract (P-EA) contained pectic polysaccharides, such as acetylated and methyl esterified homogalacturonans together with arabinogalactan, while the fraction S-EA presented phenolic acids and terpenes such as gallic acid, protocathecuic acid, liquidambaric acid, combretastatin, and atractyloside A. EA, P-EA, and S-EA showed antioxidant activity, with IC_50_ values of 4.64 µg/mL, 16.45 µg/mL, and 3.67 µg/mL, respectively. The cytotoxicity followed the sequence S-EA > EA > P-EA, demonstrating that the toxic compounds were separated from the non-toxic ones by ethanol precipitation. While the fraction S-EA is very toxic to any cell line, the fraction P-EA is a promising candidate for studies against cancer due to its high toxicity to tumoral cells and low toxicity to normal cells.

## 1. Introduction

*Liquidambar styraciflua* L., a tree that belongs to the Altingiaceae family, is an aromatic plant, dicotyledonous, hardwood and measuring 20–25 m in height. Popularly, it is known as sweet gum, alligator tree, and liquidambar. This species has wide intercontinental distribution in the south and southeast of the United States, extending to Central America and Mexico. This exotic species was acclimated in the south and southeast of Brazil because of damp soils and frosts. In traditional Chinese medicine, *L. styraciflua* is used in the treatment of gastrointestinal disorders, such as diarrhea and dysentery, coughs, and skin sores. Studies with this species have showed antioxidant capacity [1,2], hepatoprotective activity [1], acetylcholinesterase enzyme inhibition [3], antitumor action [4], anti-inflammatory and antimicrobial activities, as well as synergism with antibiotics [5,6]. Such studies evaluated mainly alcoholic extracts (hydroalcoholic and methanolic) and essential oil of leaves, bark, and stem.

According to Mancarz et al. [6], the extracts of stem and leaves of this plant showed a great anti-inflammatory response by inhibiting the hyaluronidase enzyme compared to commercial propolis extract. Their results showed that the butanolic fraction and hydroalcoholic extract of *L. styraciflua* stems were responsible for this marked anti-inflammatory effect [6]. The same extracts also demonstrated antioxidant activity when evaluated by radical scavenging assay. Extracts from bark and stem presented the higher activity, mainly the butanolic and ethyl acetate fractions that exhibited 104.4% ± 0.001 and 97.7% ± 0.001 of scavenging activity, respectively, when compared to the control containing ascorbic acid [6].

The fruits of *L. styraciflua* were also evaluated and their extracts were tested against various human cancer cell lines (PANC-1, BXPC3, AsPC-1, HCT 116, PC3, LNCaP, DU145, and A549). The 50% aqueous methanol extract, as well as a methanol extract, has demonstrated an antiproliferative effect by inhibiting mainly two prostate cancer cell lines. The aqueous methanol extract showed IC_50_ of 7.81 µg/mL, while methanolic extract presented IC_50_ of 1.85 µg/mL against PC3 cells. A similar value was obtained when the latter extract was incubated with LNCaP cells (IC_50_ 2.5 µg/mL). Lower antiproliferative activity was observed for a third prostate cancer cell line, showing IC_50_ of 31.2 µg/mL for DU145 cells [4].

Although there are some studies about *L. styraciflua* extracts, most of them were done with alcoholic extracts from the leaves and stem. Little is known about its fruits and their bioactive compounds. In this context, this work aimed to prepare *L. styraciflua* fruits extracts and investigate their chemical composition, as well as biological activities using tumoral cell lines.

## 2. Results and Discussion

The aqueous extract of *L. styraciflua* fruits (200 g) obtained by consecutive extractions showed a yield of 7.60% (g of extract/100 g of fruits). Studies with the leaves of the same species exhibited similar yield (7.355%) in the aqueous extract (Table 1), while the stems extract exhibited 1.767% [6]. The polysaccharide (P-EA) and low M*_w_* (S-EA) extracts yielded similar amounts: 3.07% and 3.32% (g of extract/100 g of fruits), respectively. The ethanol precipitation procedure allowed to separate the high molecular weight compounds, such as proteins and polysaccharides, from the low molecular weight molecules, such as phenolic compounds and other secondary metabolites [7].

The samples were evaluated for their protein and carbohydrate contents (Table 1). All fractions showed low content of proteins, and P-EA fraction presented the highest value compared to the other samples, which was expected considering that this extract concentrated the high molecular weight molecules. On the opposite side, total carbohydrates were the highest compounds observed on the three extracts, suggesting the presence of polysaccharides on P-EA fraction and oligosaccharides such as sucrose on S-EA, that were separated after ethanol precipitation of EA. Studies performed by Martin et al. [8] with bark and wood of *L. styraciflua* indicated the presence of such biomolecules. Rashed et al. [3] also observed the presence of carbohydrates in methanolic extract of the aerial parts of this species. The presence of carbohydrates in the aqueous extracts was expected since the cell wall of vegetables is composed of pectic polysaccharides, which are normally soluble in water [9]. 

Phenolic compounds are also commonly found in plants as well as in their fruits [10]. They are classified as simple phenols and polyphenols and are considered secondary metabolites of plants. More than 500 polyphenols were identified in vegetal foods, conferring nutritional and therapeutic properties of great importance [11]. Several studies have correlated the presence of such compounds with antioxidant activity because of their capacity to capture free radicals [10,12], thus preventing some diseases such as cardiovascular and degenerative disorders, as well as cancers [11]. The present study reveals that S-EA extract presented the highest level of phenolic compounds (298.4 mg/g of extract), followed by P-EA and EA extracts (Table 1). Other authors also evidenced the presence of phenolic compounds: Eid et al. [1] obtained 111.5 mg GAE/g extract for the methanol extract of the leaves of *L. styraciflua*. Extractions with other solvents provided higher contents of phenolics, such as 1614.02 ± 0.006 mg GAE/g extract with acetone and 1419.34 ± 0.033 mg GAE/g extract with ethanol extraction [6]. Wang et al. [13] obtained acetone extracts of *Liquidambar formosana* leaves, which showed 63.97 ± 0.27 mg of pyrocatechol/g extract, while the ethanolic extracts presented 76.40 ± 0.12 mg of pyrocatechol/g extract. Pyrocatechol, also known as catechol, is a phenolic compound found in plants and it presents antioxidant capacity [14]. The stem of *L. styraciflua* also presented high levels of phenolic compounds, showing 1557.57 ± 0.006 mg GAE/g extract for extraction with acetone and 875.21 ± 0.002 mg GAE/g extract for ethanol extraction [6].

### 2.1. Carbohydrates Present in L. styraciflua Extracts 

The monosaccharide content of EA was determined and showed the presence of Ara (37.8%), Glc (29.2%), Gal (8.6%), Man (6.6%), Rham (3.7%), and myo-inositol (14.1%).

The monosaccharides encountered are normally present in plant polysaccharides such as arabinogalactans and other pectic polysaccharides, while myo-inositol is a widespread metabolite observed in plants [15]. In the literature, little information was found about the content of monosaccharides in this species. However, Martin et al. [8] showed in their study the presence of xylose, mannose, glucose, and arabinose in the bark and wood of *L. styraciflua*. Neutral monosaccharides such as galactose, rhamnose, fucose, and uronic acids such as galacturonic acid were also observed in polysaccharides derived from fruits aqueous extracts [16,17,18,19].

The low M*_w_* fraction (S-EA) presented Ara (27.8%), Glc (12.3%), Man (1.1%), and myo-inositol (58.8%), indicating that arabinose and glucose could be present as oligosaccharides or being part of glycosides [20], while fraction P-EA was mainly composed of uronic acids (66.0%). The neutral sugars were Ara (23.0%), Gal (6.3%), Glc (3.3%), Man (1.4%), and Rha (trace amounts). Monosaccharide composition of the carboxyl-reduced sample showed that the uronic acid was GalA, due to the increase of the amounts of Gal from 6.3% to 64%. Other neutral sugars present in the carboxyl-reduced sample were Ara (20.5%), Glc (7.3%), Man (4.2%), and Rha (4.0%). The increase in the Rha content was expected since the glycosidic linkage between Gal and Rha in the carboxyl-reduced polysaccharide is easier to hydrolyze than that between GalA and Rha in the native polymer. This monosaccharide composition suggested the presence of pectic polysaccharides in fraction P-EA. On HPSEC analysis, the sample eluted as a polydisperse or non-uniform peak, with relative M*_w_* of 48 kDa (Figure 1).

The NMR analysis showed the presence of a methyl esterified homogalacturonan due to its typical signals in the HSQC-DEPT correlation map (Figure 2 and Appendix A). The signals were at δ 100.0/4.97 (C1-H1) from methyl esterified α-d-GalpA, δ 67.7/3.73 (C2-H2), δ 68.3/3.99 (C3-H3), δ 78.7/4.46 (C4-H4), δ 70.6/5.06 (C5-H5) from methyl esterified GalpA, and δ 52.8/3.82 from the methyl groups present at esterified C-6 (-COOCH_3_). The esterified C-6 signal was observed in the ^13^C NMR spectrum at δ 170.9. The presence of signals at δ 20.3/2.17 (CH_3_COO^−^) demonstrated that the homogalacturonan in also acetylated. This agrees with monosaccharide analysis, which showed GalA as the main sugar. Moreover, other signals were observed, such as anomeric cross peaks at δ 103.1/4.52 from β-d-Galp units, and at δ 109.1/5.26, δ 107.6/5.09, δ 107.0/5.17, and δ 106.4/5.24 from α-l-Ara*f* units from arabinan moiety. Inverted DEPT signals were seen at δ 66.7/3.81–3.89 from 5-*O*-linked α-l-Ara*f* units from arabinan main chain. Other inverted signals were at δ 61.3/3.83, and δ 61.3/3.74 from unsubstituted C-5/H-5 from α-l-Ara*f* or C-6/H-6 from β-d-Gal*p*-units. Moreover, signals of the C6/H6 from α-l-Rha units could be seen at δ 16.7/1.27 and 17.7/1.30, indicating that small amounts of rhamnogalacturonan may also be present. Altogether, these assignments are in agreement with published literature data from pectic polysaccharides [21,22,23,24,25] and indicated the presence of an acetylated and methyl esterified homogalacturonan together with arabinogalactan/type I rhamnogalacturonan in P-EA fraction.

### 2.2. Secondary Metabolites of L. styraciflua Extracts

The phytochemical composition of S-EA was determined using liquid chromatography coupled to a high-resolution mass spectrometry, operating in the negative and positive ionization polarities (Figure 3). Most of the compounds were identified on the negative mode, due to poor resolution of the positive ionization mode, with some exceptions. The components extracted from the fruits of *L. styraciflua* appeared as common phytochemicals, such as gallic acid (peak **1**, *m/z* 169.0135), protocatechic acid (peak **2**, *m/z* 153.0192), and quercetin-rhamnoside (peak **4**, 447.0909) which appeared better in the positive polarity. Such compounds produced key fragments that support their identification (Table 2).

Other compounds present in the S-EA fraction are less common and their tentative identification was assisted by ACD Labs ChemSketch (version 12.0)/ChemSpider (version 2.0.2) and PubChem. The compound **3**, at *m/z* 327.1081, was identified as a hydroxyphenylpropanoic acid glycoside, with a main fragment at *m/z* 147.0443, which corresponds to the aglycone, after the loss of the hexose moiety, with neutral loss (NL) of 180.063 atomic mass units (a.m.u.). Similar compound was found in other plants, such as *Adansonia digitata* L., being described as dihydro melilotoside–dihydro coumaroyl *O*-hexoside [26]. The compound **5** appeared at *m/z* 483.1978 and 493.2280: these is an uncommon ionization profile, being identified as [M + Cl]^−^ and [M + HCOO]^−^, respectively, based on the isotopologue distribution of chlorinated ion, that exhibited an abundant M + 2 isotopologue (*m/z* 485.19). This compound was tentatively identified as Atractyloside A, a diterpene glycoside, formerly described in the *Atractylodes lancea* (Thunb.) and other plants [27,28]. 

The compounds **6** and **7**, at *m/z* 327.2181 and 329.2335, respectively, were possibly related to terpenes with a double bond difference, however they were not identified. The compound **8**, at *m/z* 333.1346, was tentatively identified as combretastatin, a compound from group of natural phenols isolated from the root bark or stem of *Combretum caffrum* [29]. Three isoprenoids (triterpenic acids) were found in the *L. styraciflua* fruits extract: the peak **9**, at *m/z* 469.3322 [(C_30_H_45_O_4_)^−^], was consistent with the structure of 6β-hydroxy betunolic acid, previously observed on cuticles of *L. styraciflua* [30]. The two other triterpenic acids (compounds **10** and **11**) were found as isomers, at *m/z* 453.336 [(C_30_H_45_O_3_)^−^]. Such compounds were consistent with isomers of betulonic acid, also named as liquidambaric acid, a pentacyclic triterpene formerly isolated from ethyl acetate extract of *L. formosana*, as described by Yan et al. [31].

### 2.3. Antioxidant and Antitumoral Effects of L. styraciflua Extracts 

Plants are the biggest producers of organic matter and provide a wide variety of naturally bioactive compounds. Like the stem and leaves, the fruits can also synthesize macromolecules and secondary metabolites that have important biological properties, such as antioxidant activity. The radical scavenging assay (Figure 4) showed that the antioxidant capacity of S-EA (IC_50_: 3.67 µg/mL) was slightly higher than that observed on the crude extract (EA) (IC_50_: 4.64 µg/mL), followed by the polysaccharide fraction P-EA (IC_50_: 16.45 µg/mL). It was also observed that the three fractions showed an antioxidant effect like ascorbic acid at concentration of 100 µg/mL. According to Lingbeck et al. [32], three species of the genus *Liquidambar* sp. presented antioxidant property: *L. styraciflua*, *L. formosana*, and *L. orientalis*. Studies performed with extracts obtained from barks, stems, and leaves of the same species revealed antioxidant properties, with emphasis on the ethyl acetate and butanol fractions of the bark and stem that showed greater activity in relation to ascorbic acid [2].

Reactive oxygen species (ROS) are involved with a biochemical imbalance that can damage biological macromolecules and initiate some diseases, such as atherosclerosis, degenerative and neurological diseases, and cancer. The search of plants with antioxidant capacity, that can be used as food, tea, food supplement, or medicine, is of great importance since the amounts of antioxidants produced by the body are often insufficient to prevent pathologies or reduce damage caused by ROS [33].

The presence of phenolic acids and terpenes on S-EA fraction corroborate with literature data about the antioxidant capacity of secondary metabolites, showing the greater effect of this fraction.

Phytochemicals are vastly explored as cancer therapy and almost 70% of anticancer drugs approved between 1980 and 2002 were derived from natural products or based on knowledge of such compounds. Some examples are camptothecin, the terpene paclitaxel, and derivative compounds from the lignan class such as etoposide and teniposide [34]. Phytotherapy can be based on plant extracts as well as on isolated small molecules or macromolecules. As many plant extracts may present toxic compounds, the first step to be verified is their toxicity after incubating them with cells, and then associate the effect of specific preparation to its chemical composition.

Cell viability can be measured by different methods, which may evaluate membrane integrity, normal adherence, and normal metabolic activity. For this purpose, MTT method was used to determine the cell viability of three different cell lines to verify the toxicity of *L. styraciflua* extracts [35]. The cytotoxic properties of the three preparations (EA, P-EA, S-EA) were investigated in different incubation periods using tumoral hepatic (HepG2), normal and tumoral breast (MCF7 and MCF10A, respectively) cells. As shown in Figure 5, it is possible to observe that EA and S-EA extracts follow similar patterns, presenting high toxicity (<40% viability) to HepG2 cells at 1000 µg/mL, while P-EA fraction showed no toxicity even at high concentrations. This result demonstrates that toxic compounds present in the aqueous extract were separated from the non-toxic ones after the ethanol precipitation, remaining in the S-EA fraction, while P-EA accumulated only polysaccharides that usually do not cause harm to cells [36].

As many plant compounds, even the non-toxic molecules, may present biological activities, including anticancer effects; a tumoral breast cancer cell line was used to verify such properties of EA, P-EA, and S-EA extracts [34,36,37]. The aqueous extract (EA) was not toxic to MCF7 cells up to 300 µg/mL after 24 h and 48 h of incubation (Figure 6). While S-EA presented high toxicity (≥40% viability reduction) at the same concentration and same incubation periods, with IC_50_ values of 261.1 µg/mL (24 h) and 283.3 µg/mL (48 h). P-EA fraction was not toxic to the tumoral breast cells after 24 h of incubation. However, after 48 h of incubation, this fraction promoted a reduction of ~25% on cell viability, at 1000 µg/mL (Figure 6), which is not considered toxic by ISO 10993-5 (2009) [38]. As previously discussed, P-EA is mainly composed of homogalacturonan and arabinogalactan, which are not expected to present toxicity [39]. However, extracts and polysaccharides from different fruits were observed to induce apoptosis and oxidative stress, inhibit proliferation through cell cycle arrest, and inhibit angiogenesis [40]. The mechanism of action of such molecules were not totally elucidated, and they usually show effect at concentrations higher than 100 µg/mL [40]. Polysaccharides usually present important biological activities such as antioxidant [41], anti-inflammatory [42], and immune-modulatory effects [39]. Such properties combined can also lead to an indirect antitumor activity, which is another good reason to investigate plant polysaccharides, their chemical structure, and biological action. Furthermore, tumoral cells are usually more resistant to drugs due to different mechanisms, which boosts researchers to find for new drugs that affect mainly tumoral cells and do not harm normal cells. It was observed that arabinogalactans of low M*_w_* and other saccharides containing β-d-Gal*p* units may act as galectin-inhibitors. Galectins are potential targets for cancer therapy because of their relationship with important physiological and pathological functions, such as immune response, cell migration and signaling, inflammation, fibrosis, or cancer development and progression [43]. However, some authors have demonstrated that pectic polysaccharides bind only poorly to the carbohydrate binding sites of different galectins [43]. The results of this study showed a slight reduction on cell viability of tumoral cells after 48 h, only on the highest concentration of P-EA, which contain arabinogalactans. One possible mechanism of action for this extract is the inhibition of proliferation by binding to galectin.

A longer incubation period (Figure 6) was evaluated with the three extracts (EA, P-EA, and S-EA) on MCF7 cells (300–1000 µg/mL) and again a similar pattern was observed for EA and S-EA, showing a high toxicity of both extracts with IC_50_ of 328.4 µg/mL for EA and 502.3 µg/mL for S-EA. The low M*_w_* fraction (S-EA) is mainly composed of phenolic acids and terpenes, which are synthesized by plants as defense against pathogens and other predators, as well as signaling compounds for insect pollination. Aside from that, such metabolites are known to possess toxicity among other pharmacological features that could be employed as medicine [34]. The challenge to study plants extracts and their biological applications is to identify phytochemicals present in the preparations, that are usually a mixture of flavonoids, terpenes, polyphenols, and a number of isomers. In the present study, some secondary metabolites from S-EA fraction were identified such as liquidambaric acids, which are known to present antitumoral, antiviral, and anti-malarial activities [44,45,46]. Another compound detected *in L. styraciflua* fruits was combretastatin, which is known as a potent anti-tubulin agent, acting as antiproliferative and antitumoral [29]. The triterpenoid 6β-hydroxy betunolic acid was reported to present antifungal and antibacterial activity; however, its effect on human cells were not mentioned [47]. Moreover, the diterpene atractyloside A detected in S-EA fraction was previously isolated from other plants and it is highly toxic, causing acute hepatic or renal pathology, through the inhibition of the mitochondrial ADP transporter [48]. This compound may be the principal responsible for the toxic effects of EA and S-EA observed on HepG2 and MCF7 cells (Figure 5 and Figure 6). After a longer period of incubation, P-EA fraction presented a moderate toxicity to the tumoral cells and gradually reduced their viability by ~30%, 40% and ~50% at 500, 750, and 1000 µg/mL, respectively (Figure 6). The increase in toxicity after a longer period of incubation may indicate that galactose present in P-EA fraction increased chances of binding to galectin, consequently promoting ~50% of viability reduction at 1000 µg/mL (Figure 6H). However other mechanisms could be involved, and more experiments should be performed to confirm this hypothesis. Other plant polysaccharides promoted cell cycle arrest, induced apoptosis, and inhibited cell growth of MCF7 cells, but the exact mechanism of action of such plant compounds were not defined [40].

Antitumoral activity of fruit extracts from the same species was previously observed [4], when the aqueous extract and its fractionation with methanol (20, 50, 100%) was tested in several tumor cells. Two different prostatic adenocarcinoma cell lines (PC3 and LNCaP) showed antiproliferative activity when treated with the 100% methanol fraction, indicating a high antitumor activity (IC_50_: 1.85 µg/mL and IC_50_: 2.5 µg/mL, respectively). Although with lower activity, this same fraction also inhibited the growth of HCTI16 (colon carcinoma), Panc-1 (human pancreatic carcinoma), and DU145 (prostatic adenocarcinoma) cells. Aside from interesting effect against tumoral cell lines being observed, the absence of a non-tumorigenic cell line test is a point to be evaluated about the toxicity of the methanol fraction. Furthermore, the methanol fraction prepared by Liu et al. [4] may possibly contain the same toxic and antitumoral compounds observed in the present study (fraction S-EA) due to similarity of solvents utilized.

A successful anticancer drug is that one able to kill tumor cells without harming the normal tissue, and due to the well-known resistance developed by tumoral cells against numerous drugs, this problem remains without solution [34]. In the present study, it was observed that S-EA presented the greater toxicity to tumoral cells lines HepG2 (Figure 5) and MCF7 (Figure 6). While the P-EA extract, mostly composed by homogalacturonan and arabinogalactan showed no toxicity to both cells after 24 h of incubation; presented a slight reduction in viability (by ~30%) of MCF7 cells at the highest concentration (1000 µg/mL) after 48 h of incubation; and showed moderate toxicity to MCF7 cells after 72 h of incubation. When the extracts produced from *L. styraciflua* fruits were tested on normal breast cells (MCF10A) as comparison (Figure 7), it was observed that P-EA did not affect the viability of cells, even at the higher concentration (1000 µg/mL), after 72 h of incubation. This interesting result demonstrates the potential of P-EA as a candidate for anticancer drug studies. Another homogalacturonan rich-pectin, isolated from ginseng showed inhibition of tumor cell proliferation and cell cycle arrest at G2/M phase of HT-29 cells (human colon cancer) [36].

The antitumor activity of polysaccharides is usually a consequence of their immune-stimulating property, due to stimulation of macrophages and NK cells to produce pro-inflammatory cytokines [36]. However, in the present study, it was observed that MCF7 cells were directly affected by *L. styraciflua* polysaccharides, by reducing their mitochondrial activity, observed on MTT assay, effect that was not observed on normal breast cells (Figure 6 and Figure 7). The human organism is constantly undergoing internal and external stimuli, which can cause physiological or pathological cell death, either by apoptosis or necrosis. These two pathways of cell death occur independently and depend on the conditions, intensity, and duration of the stimuli to which the organism is submitted [49]. In this context, P-EA and S-EA extracts were incubated with MCF7 cells to determine which cell death pathway is stimulated. For this experiment, the cell cycle was evaluated on MCF7 cells treated only at concentrations of extract that maintain cells with 50% of viability or more (P-EA: 500–1000 µg/mL; S-EA: 100–300 µg/mL). The results demonstrated that P-EA and S-EA (Figure 8A,B) did not produce any arrest on MCF7 cells cycle at the evaluated concentrations, when they were incubated for 72 h.

Considering that only S-EA presented prominent reduction of cell viability on MTT assay, an apoptosis and necrosis assessment was performed after incubating cells with this fraction, and it was observed that apoptotic cells increased while viable cells decreased at raising concentrations of S-EA (Figure 8C), following the same pattern observed on MTT assay (Figure 6). Apoptosis started at 500 µg/mL, culminating with a high number of cells in late apoptosis at 1000 µg/mL. The atractyloside A present in S-EA is known to induce apoptosis at low doses in vitro and besides destruct the tubulin network, it can also derange mitochondrial morphology and inhibit cell division [48]. Phytochemicals able to induce apoptotic cell death, especially against tumoral cells, are considered promising oncologic drugs due to the maintenance of the membrane integrity and absence of inflammatory reaction.

The toxic effect observed for P-EA and S-EA extracts demonstrates an interesting property of this plant, that is composed of compounds with different levels of toxicity, which is probably related to different mechanisms of action: while arabinogalactans and homogalacturonans present in P-EA are macromolecules that probably interact with membrane receptors of cells [36], the secondary metabolites present in S-EA may possibly be internalized by the cells, affecting different paths [48]. More studies are necessary to determine which compound is the responsible for the toxicity observed, as well as which path is being altered, and to understand the pharmacological potential of *L. styraciflua* extracts.

## 3. Material and Methods

### 3.1. Plant Material

The fruits of *L. styraciflua* L. were collected at Fazenda Experimental Canguiri (CEEx), located in Pinhais, Paraná State, in South Brazil (latitude: 25°23′12.3″ S, longitude: 49°07′33.2″ W, altitude: 920 m), during the autumn of 2019. The plant was identified, and a voucher specimen (No. 91728) was deposited in the UPCB Herbarium of Botany Department of Federal University of Paraná. The studied species was registered on Sistema Nacional de Gestão do Patrimônio Genético e do Conhecimento Tradicional Associado (SISGEN), number A5023C7.

### 3.2. Extraction Procedures

The fruits were dried in the dark at 25 °C, for 14 days. Afterwards, they were milled and submitted to successive aqueous extractions, at 100 °C for 4 h (×3; 1000 mL each), under reflux (Figure 9). The aqueous extract (EA) was concentrated under reduced pressure, at 60 °C, and the polysaccharide extract (P-EA) was obtained after precipitation with ethanol (3:1; *v/v*) and centrifugation, at 10,000 rpm, at 10 °C, for 20 min. The precipitate was dialyzed against tap water, on an open system, for 24 h, using dialysis tube of M*_w_* 3.5 kDa cut-off, concentrated under reduced pressure and freeze-dried. The supernatant (S-EA), which contained small molecular weight compounds was evaporated to a small volume, under reduced pressure, at 50 °C, freeze-dried.

### 3.3. Determination of Phenolic Compounds, Proteins, Carbohydrates, and Uronic Acid

The phenolic compounds were analyzed by the Folin–Ciocâlteu method [50]. For standard curve, gallic acid (GA) was prepared in the concentrations (12.5–50 µg/mL), and the absorbance was measured at 595 nm. The results were expressed as mg GA equivalents/g extract. The proteins were performed using the Coomassie^®^ Brilliant Blue G-250 reagent, according to the methodology described by [51]. Calibration curve was prepared with bovine serum albumin (BSA) (0.1–1.4 mg/mL) and absorbance was measured at 595 nm. The results were expressed as mg BSA equivalents/g extract. The determination of carbohydrates was performed using the phenol–sulfuric acid method [52]. A solution of D-glucose (Glc) was used to obtain the standard curve (0.032–0.8 mg/mL), and the absorbance was measured at 490 nm. The results were expressed as mg Glc equivalents/g extract. Uronic acid contents were determined using the modified sulfamate/m-hydroxybiphenyl method [53] employing galacturonic acid as standard.

### 3.4. Reduction of Uronic Acids

Uronic acids present in polysaccharides fraction P-EA were reduced using carboxyl reduction method of [54]. The procedure was carried out using the 1-ethyl-3-(3-dimethylaminopropyl)-carbodiimide to give a carboxy-reduced fraction. Sodium borohydride was used as reducing agent.

### 3.5. Monosaccharide Analysis

The aqueous (EA), polysaccharide (P-EA), and low Mw (S-EA) extracts were hydrolyzed with 1 M trifluoroacetic acid (TFA) at 100 °C for 14 h, and the solution was evaporated with nitrogen flow. Afterwards, the samples were sequentially washed with acetone (1 mL) and ethanol (1 mL), followed by evaporation. The samples were reduced with sodium borohydride solution (2 mg/mL, pH 9.0), at room temperature for 14 h, and later they were treated with cationic resin (20 mg) until reaching pH 5.0. The solution was evaporated to dryness and the resulting boric acid was removed by addition of methanol followed by evaporation (3×). The acetylation process was performed using acetic anhydride-pyridine (200 µL; *v/v*) at room temperature for 14 h. The resulting alditol acetates were extracted with acetone and analyzed by GC–MS (Shimadzu) using rtx5-ms column (30 m × 0.25 mm) programmed to raise the temperature from 50 °C to 250 °C, at a rate of 10 °C/min.

### 3.6. Liquid Chromatography-Mass Spectrometry (LC-MS) Analysis

The ethanolic supernatant fraction (S-EA) was analyzed by ultra-performance liquid chromatography (UPLC™, Acquity, Waters), incorporated with a binary pump, sample manager, and column oven, coupled to a high-resolution mass spectrometer (HR-MS) with quadrupole time-of-flight (Xevo G2-S, Waters Co., Milford, USA) with an electrospray ionization interface. A BEH-C18 column (Waters), with 100 mm × 2.1 mm and 1.7 µm particle size was used.

The chromatography was developed at a flow rate of 0.4 mL/min, with a mobile phase consisting of H_2_O and acetonitrile, containing 0.1% formic acid (*v/v*). The separation was performed with a linear gradient, increasing acetonitrile from 5 to 50% in 7 min, to 95% at 10 min, held to 16 min, returning to the initial condition in 18 min, with 2 min else for column re-equilibration. The samples (1 mg/mL) were prepared in MeOH, held at room temperature (22 °C), and 5 µL was injected in each analysis. Detection was provided by HR-MS. High purity nitrogen used as cone gas and desolvation was produced by a nitrogen generator from Peak Scientific Instruments (Chicago, IL, USA). The collision gas used was argon with purity > 99.998% from White Martins Praxair Inc. (Curitiba, Brazil). The source temperature was 150 °C, capillary at 3 kV, collision energy ramp of 30–50 eV, desolvation temperature of 350 °C. All data were acquired from 100 to 1500 m/z in centroid mode, using MassLynx™ NT4.1 software (Waters Co., Milford, CT, USA). For the mass accuracy, leucine encephalin was used as internal calibrants.

### 3.7. Determination of Homogeneity and Relative Molecular Weight

The homogeneity and relative molecular weight (M*_w_*) of fraction P-EA was evaluated by high performance steric exclusion chromatography (HPSEC), with a Waters 2410 differential refractometer as equipment for detection. A series of four columns (Ultrahydrogel 2000, 500, 250, 120, Waters) was used. The eluent was 0.1 M aq. NaNO_2_ containing 200 ppm aq. NaN_3_ at 0.6 mL/min. The sample, previously filtered through a membrane (0.22 µm, Millipore Sigma, Burlington, USA), was injected at a concentration of 1 mg/mL. To obtain the relative Mw, standard dextrans (72.2 kDa, 40.2 kDa, 17.2 kDa, 9.4 kDa, and 5 kDa, from Millipore Sigma, Burlington, USA) were employed to obtain the calibration curve. The relative Mw of the sample was calculated according to the calibration curve.

### 3.8. Nuclear Magnetic Resonance Spectroscopy

The NMR experiments (Heteronuclear Single Quantum Correlation—Distortionless Enhancement by Polarization Transfer: ^1^H, ^13^C, and HSQC-DEPT) were obtained using a 400 MHz Bruker spectrometer model Avance III with a 5 mm inverse probe. The polysaccharide extract (P-EA, 50 mg) was dissolved in D_2_O and the analyses were performed at 70 °C. Chemical shifts (δ) were expressed in ppm relative to the solvent ^13^C (δ 30.2) and ^1^H (δ 2.22) resonances. The NMR chemical shifts were assigned according to HSQC-DEPT experiments and literature data.

### 3.9. Scavenging Activity of L. styraciflua Extracts Measured by DPPH Assay

The antioxidant activity was determined following the method described by [55], with some modifications. All samples (EA, P-EA and S-EA) were evaluated at concentrations of 1–100 µg/mL. The analytical curve was obtained with a DPPH solution (0–60 µM). The ascorbic acid (50 μg/mL) was used as positive control, while ultrapure water (EA and S-EA) or 1% dimethyl sulfoxide solution (P-EA) were used as vehicle controls. The absorbance was measured at 517 nm using a microplate reader (Biotek^®^, Epoch, Agilent, Santa Clara, USA).

### 3.10. Cell Culture

Three different cell lines were used in this study: human breast cancer cell line (MCF7), normal human breast epithelial cell line (MCF10A), and human hepatocellular carcinoma cell line (HepG2). The cells were kept under the same conditions: they were maintained in DMEM F-12 culture medium supplemented with 10% fetal bovine serum and 1% penicillin/streptomycin (P/S) (Millipore Sigma, Burlington, USA). Cells were incubated in a humidified atmosphere containing 5% CO_2_ at 37 °C and the medium was renewed twice a week.

### 3.11. Cell Treatment with L. styraciflua Extracts

For the cell viability assay, MCF7, MCF10A, and HepG2 (1 × 10^4^ cells/well) were plated in 96-well plates. The cells were incubated under 5% CO_2_ at 37 °C for 24 h. After this period, the medium was removed and of the extracts EA, P-EA, and S-EA (1, 3, 10, 30, 100, 300, and 1000 µg/mL) were added to the cells. The plates were incubated under the same conditions, for 21 h and 45 h and MTT assay was performed to verify cytotoxicity. For the 72 h treatment, MCF7 cells were plated in 48-well plates, while the MCF10A cells were plated in 96-well plates. The cells were incubated under the same conditions. After that, the medium was removed and the extracts EA, P-EA, and S-EA (300, 500, 750, and 1000 µg/mL) were added to the cells. The plates were incubated again for 69 h and MTT assay was performed to verify cell viability.

### 3.12. Cell Viability Assay

The MTT assay [56] was performed to verify cell viability. The medium with the samples was removed from the wells and fresh medium containing MTT (0.45 mg/mL) was added to each well. The plates were incubated for 3 h under 5% CO_2_ at 37 °C. Afterwards, 200 µL of DMSO for 24 and 48 h plates, and 450 µL of DMSO for 72 h plates were added to each well to solubilize the formazan crystals produced by living cells, and the plates were placed on a shaker for 15 min for complete solubilization. Later, the plates were read at 595 nm in a microplate reader (Biotek^®^, Epoch, Agilent, Santa Clara, USA). The change of yellow color to purple color due to the formation of formazan crystals shows the presence of viable cells. The absorbance obtained from untreated control cells was used to check the toxicity of vehicle, and the vehicle was considered as 100% cell viability. Reduction of cell viability by more than 30% was considered as a cytotoxic effect, according to ISO 10993-5 (2009).

### 3.13. Cell Cycle

Cell cycle phases can be assessed through DNA content after labeling with 7AAD by flow cytometry. For this analysis, MCF7 cells were plated at a density of 4 × 10^5^ cells/well in a 6-well plate. Cells were incubated for 72 h with P-EA (500, 750 and 1000 µg/mL) and S-EA (100 and 300 µg/mL). After this period, the cells were washed with PBS and resuspended in ice-cold 70% ethanol, followed by incubation at −4 °C for 24 h for fixation. After that, an aliquot of PBS + 2% BSA was added to inhibit possible unspecific binding with 7AAD. Then, the supernatant was removed, and the cells were resuspended in 100 µg/mL RNAse (0.1% Triton-X in PBS) and labeled with 7AAD. DNA analysis was performed by flow cytometry and each phase of the cycle was determined using the FlowJo X software, version 9.

### 3.14. Apoptosis and Necrosis Assessment

Annexin V is a protein that binds to phosphatidylserine, which is a phospholipid present in cell membranes that acts in the regulation of apoptosis and interferes in several cell signaling pathways. Annexin V marks cells that are in an early stage of apoptosis, when the membrane remains intact. 7AAD enters the cell only when membrane integrity is lost, which indicates late apoptosis or even necrosis. For evaluation of apoptosis and necrosis, MCF7 cells were plated at a density of 4 × 10^5^ cells/well in a 6-well plate. Cells were incubated for 72 h with S-EA (500, 750, and 1000 µg/mL). After this period, the cells were washed with ice-cold PBS and then removed. Cells were resuspended in binding buffer (1x) and incubated for 15 min at 4 °C with Annexin-V-FITC and 7AAD. Analyses were performed by flow cytometry and the population of viable cells, early apoptosis, late apoptosis and necrosis were observed using Flowing Software 2.5.0.

### 3.15. Statistical Analysis

Data were analyzed by the software GraphPad Prism (GraphPad Software, USA), and presented as median. Statistical analysis was performed using a non-parametric *t*-test with unpaired *t*-test. The significant difference between groups was considered when *p* < 0.05.

## 4. Conclusions

The results obtained in this study show an interesting antioxidant activity of the three fractions tested, especially the fraction containing the higher content of phenolic acids and terpenes (S-EA). This plant may have great potential for the treatment of pathologies directly related to the imbalance and damage caused by reactive oxygen species. Comparing the cytotoxic effects of the fractions, S-EA and EA follow similar patterns in the cells tested, which concludes that the toxic compounds present in the EA remained the low M*_w_* fraction (S-EA). The present study demonstrated an interesting result of the P-EA extract as it exhibited moderate toxicity to tumor cells, which was not observed for normal cells after 72 h. In this way, this extract may become a promising candidate for studies against cancer.

## Figures and Tables

**Figure 1 molecules-28-00360-f001:**
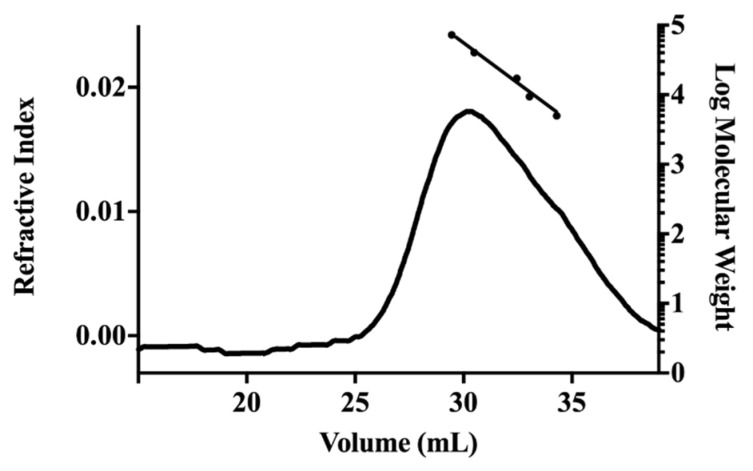
HPSEC elution profile of fraction P-EA. Refractive index detector. Elution volume of dextran standards of molecular weight 72.2 kDa, 40.2 kDa, 17.2 kDa, 9.4 kDa, and 5 kDa (left to right) were employed to construct the calibration curve.

**Figure 2 molecules-28-00360-f002:**
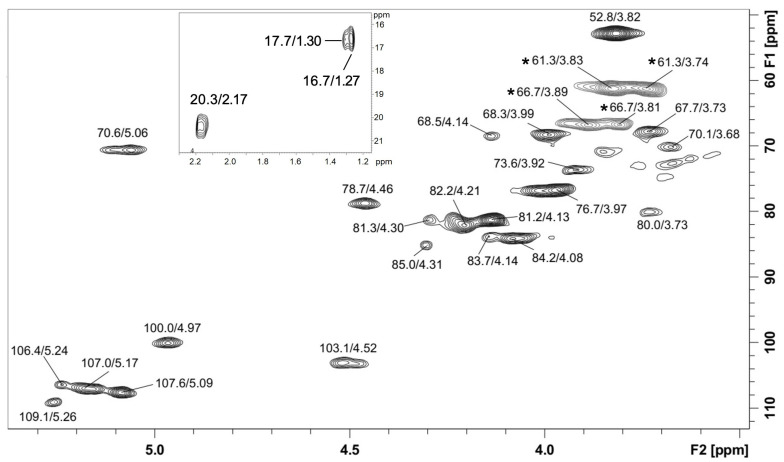
HSQC-DEPT correlation map of P-EA fraction in D_2_O at 70 °C, the chemical shifts are expressed as δ (ppm). Inverted signals in DEPT experiment are marked with *.

**Figure 3 molecules-28-00360-f003:**
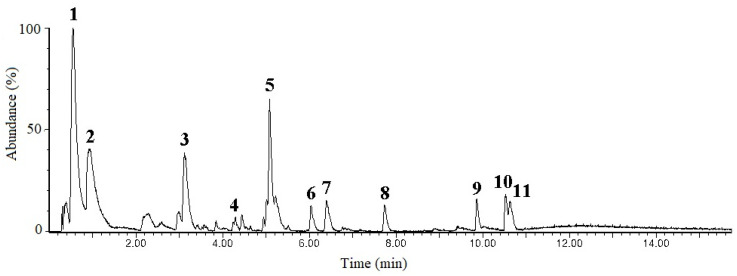
Chromatographic profile of S-EA fraction obtained on LC-MS analysis.

**Figure 4 molecules-28-00360-f004:**
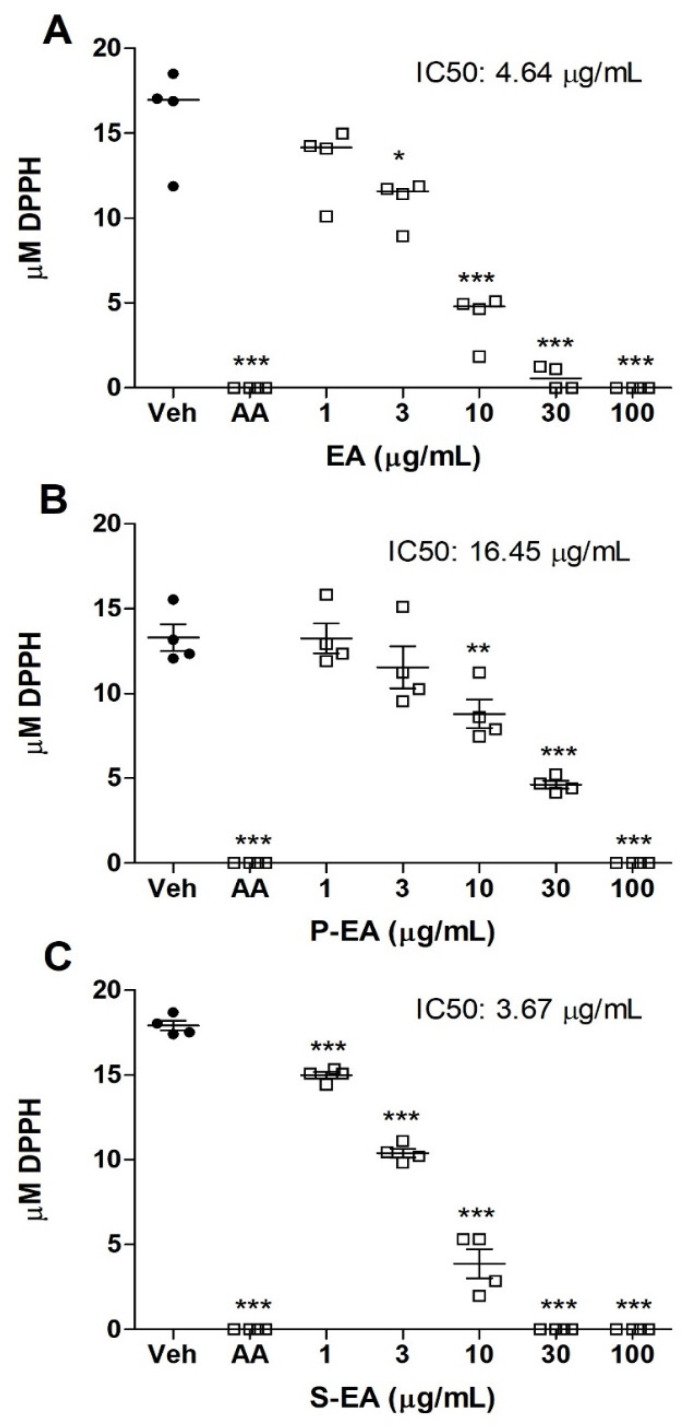
Analysis of the antioxidant effect of *L. styraciflua* fruits extracts on DPPH radical scavenging. Aqueous (EA) (**A**), polysaccharide (P-EA) (**B**), and Low M*_w_* (S-EA) (**C**) extracts were tested at 1–100 µg/mL or 0: negative control (vehicle). AA: positive control (ascorbic acid, 50 µg/mL). Statistical analyses were performed by unpaired *t*-test. Results represent the median of three independent experiments (*n* = 4) and * *p* < 0.05; ** *p* < 0.01; *** *p* < 0.001.

**Figure 5 molecules-28-00360-f005:**
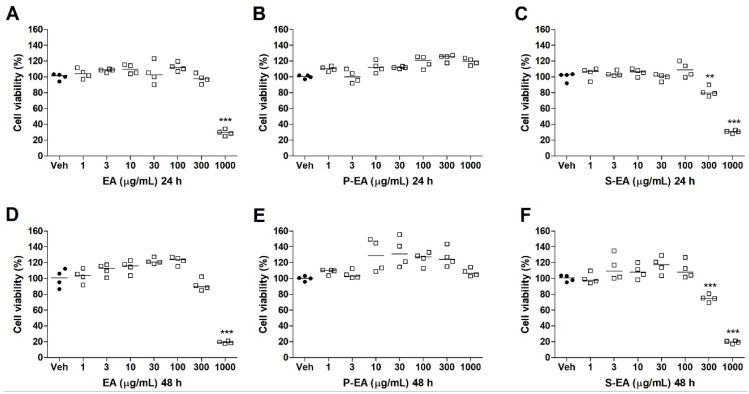
Cell viability of HepG2 cell line treated with EA (**A**,**D**), P-EA (**B**,**E**), and S-EA (**C**,**F**). Cells were incubated with the extracts (1, 3, 10, 30, 100, 300, or 1000 µg/mL) for 24 h and 48 h. Statistical analyses were performed by unpaired *t*-test. The results represent the median of two independent experiments (*n* = 4). ** *p* < 0.01; *** *p* < 0.001 compared to vehicle group (control group).

**Figure 6 molecules-28-00360-f006:**
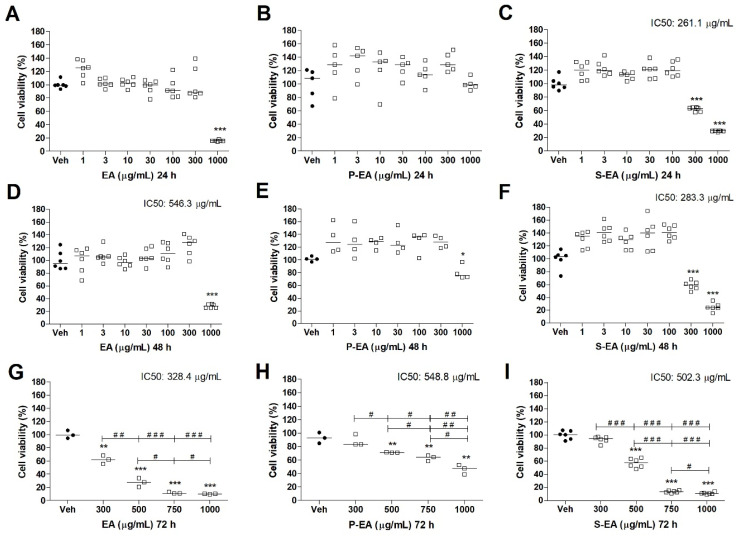
Cell viability determined by MTT assay of MCF7 cell line treated with EA (**A**,**D**,**G**), P-EA (**B**,**E**,**H**), and S-EA (**C**,**F**,**I**). Cells were incubated with the extracts (1, 3, 10, 30, 100, 300, or 1000 µg/mL) for 24 h and 48 h, and 300, 500, 750, or 1000 µg/mL for 72 h. Statistical analyses were performed by unpaired *t*-test. The results represent the median of three independent experiments (*n* = 3–6). * *p* < 0.05; ** *p* < 0.01; *** *p* < 0.001 compared to vehicle group (control group). # *p* < 0.05; ## *p* < 0.01; ### *p* < 0.001 comparison with indicated pairs.

**Figure 7 molecules-28-00360-f007:**
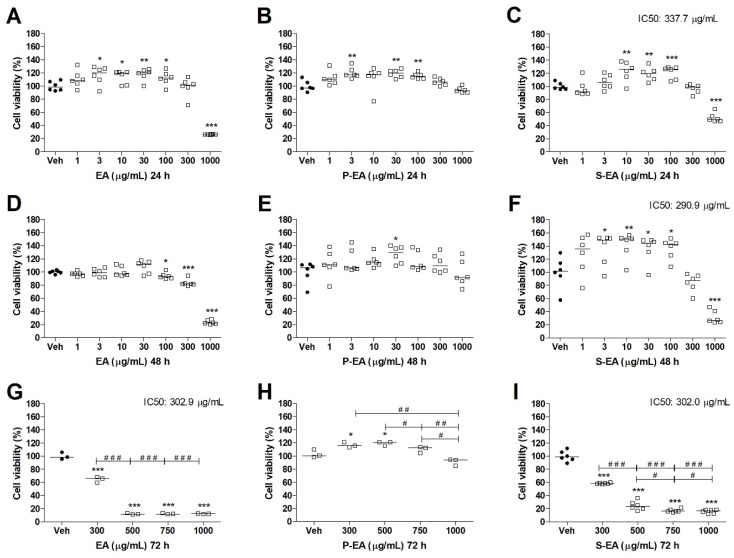
Cell viability determined by MTT assay of MCF10A cell line treated with EA (**A**,**D**,**G**), P-EA (**B**,**E**,**H**), and S-EA (**C**,**F**,**I**). Cells were incubated with the extracts (1, 3, 10, 30, 100, 300, or 1000 µg/mL) for 24 h and 48 h, and 300, 500, 750, or 1000 µg/mL for 72 h. Statistical analyses were performed by unpaired *t*-test. The results represent the median of three independent experiments (*n* = 3–6). * *p* < 0.05; ** *p* < 0.01; *** *p* < 0.001 compared to vehicle group (control group). # *p* < 0.05; ## *p* < 0.01; ### *p* < 0.001 comparison with indicated pairs.

**Figure 8 molecules-28-00360-f008:**
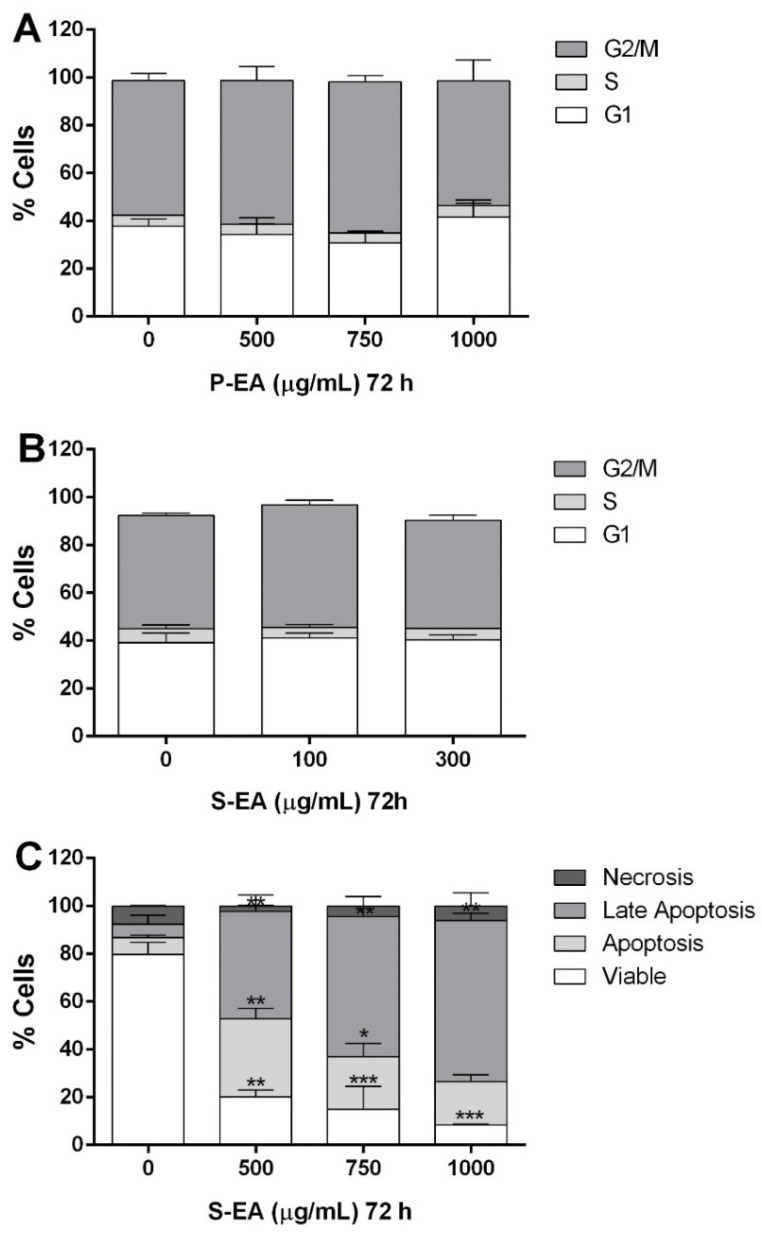
Analysis of the cell cycle progression of MCF7 cells after incubation with P-EA fraction (**A**) and S-EA fraction (**B**), and apoptosis induction after incubating with S-EA (**C**), using flow cytometry. For cell cycle, the cells were treated with P-EA (500, 750 or 1000 µg/mL) and S-EA (100 or 300 µg/mL) for 72 h. For analysis of apoptosis, the cells were treated with S-EA (500, 750 or 1000 µg/mL) for 72 h. Statistical analyses were performed by unpaired *t*-tests. The results represent the mean ± SD of two independent experiments (*n* = 3). * *p* < 0.05; ** *p* < 0.01; *** *p* < 0.001 compared to vehicle group (control group).

**Figure 9 molecules-28-00360-f009:**
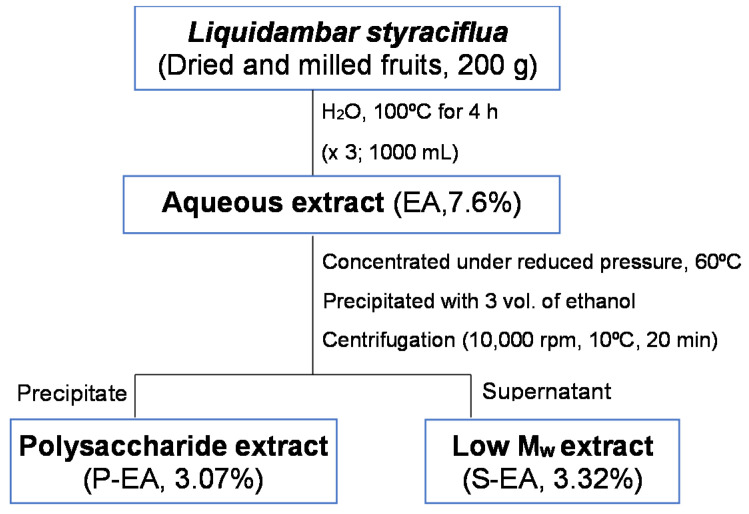
Schematic representation of the *L. styraciflua* fruits extraction.

**Table 1 molecules-28-00360-t001:** Extraction yield, proteins, carbohydrates, and phenolic compounds content of the fractions obtained from *L. styraciflua* fruits.

Fractions	Extract Yield (%)	Proteins (mg eq. BSA)/g Extract)	Carbohydrates (mg eq. Glc)/g Extract)	Phenolic Compounds (mg eq. GA)/g Extract)
**EA**	7.60	14.2 ± 0.06	250.0 ± 0.01	100.7 ± 0.85
**P-EA**	3.07	33.0 ± 0.07	250.0 ± 0.03	113.1 ± 3.00
**S-EA**	3.32	Tr.	200.0 ± 0.01	298.4 ± 8.12

Tr.: Traces. Data represent mean ± SD.

**Table 2 molecules-28-00360-t002:** Putative identification of the compounds observed on S-EA fraction obtained on LC-MS analysis.

Peak	Rt (min)	MS^1^	MS^2^	Tentative Identification	References
**1**	0.55	169.0135	125.0240	Gallic acid	-
**2**	0.94	153.0192	109.0297	Protocatechuic acid	-
**3**	3.12	327.1081	147.0443	Hydroxyphenyl propanoic acid glycoside	[26]
**4**	4.33	447.0909	300.0308/301.0387	Quercetin-rhamnoside	-
**5**	5.08	483.1978/493.2280	315.1793	Atractyloside A	[27,28]
**6**	6.04	327.2181	--	N.I.	-
**7**	6.41	329.2335	--	N.I.	-
**8**	7.74	333.1346	--	Combretastatin	[29]
**9**	9.87	469.3322	--	6β-hydroxy betunolic acid	[30]
**10**	10.53	453.3367	--	Liquidambaric acid (isomer 1)	[31]
**11**	10.64	453.3369	--	Liquidambaric acid (isomer 2)	[31]

N.I. = not identified.

## Data Availability

Data sharing not applicable.

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
