# Peer review of "Chemical Evaluation of Liquidambar styraciflua L. Fruits Extracts and Their Potential as Anticancer Drugs"

_molecules, 2023, doi:10.3390/molecules28010360_

Round 1
Reviewer 1 Report
The authors have conducted studies to investigate the chemical composition and biological potential of extracts obtained from the fruits of Liquidambar styraciflua L. An aqueous extract (EA) was prepared and mixed with ethanol to give precipitable (P-EA) and soluble (S-EA) fractions which comprised of proteins, phenolic compounds, and carbohydrates in different proportions. P-EA contained pectic polysaccharides, such as acetylated and methyl esterified homogalacturonans together with arabinogalactan while S-EA contained phenolic acids and terpenes such as gallic acid, protocathecuic acid, liquidambaric acid, combretastatin and atractyloside A. EA, P-EA,and S-EA had antioxidant activity. They report that S-EA was toxic to normal and tumour cells whereas P-EA had low toxicity for normal cells in culture but high toxicity against tumour cells and was therefore promising candidate for studies against cancer.
However, the comparisons between the effects of EA, P-EA, S-EA on cells are not alike. Studies with HepG2 cells were done over 24 and 48 hours whereas the study with MCF7 tumoral cells were done at 24, 48 and 72 hours with IC50 measured at 72 hours. What is to say that similar toxicities would not have been evident with HepG2 cells after 72 hours? Also, the authors need to have tested against a range of tumour cells rather than only one to make a robust statement that the extracts had anti-tumour actions. With one cell type the findings could be by chance.
Proteins, carbohydrates, and phenolic compounds comprise at best fifty percent of the material in EA, P-EA, S-EA extracts. What is the rest?
What information presented is novel? What is different about use of the fruit instead of bark and stem as a source of bioactive compounds? Much of presented data seems to be confirmatory of what is found in bark and stem. Are they more readily available from fruit, are the yields better, or do the extracts have unexpected properties about effects on cells.
As it stands EA, P-EA, S-EA extracts each are mixtures of ill-defined factors that can influence the metabolism of tumour cells in culture, and P-EA is touted as having most potential. However, there are hundreds if not thousands of plant extracts which show similar properties. So, what makes this so different. The authors need to show that P-EA can be effective in a cancer model in vivo or show mechanisms by which P-EA can block growth of tumour cells in vitro which might indicate potential routes of action in vivo.
Ln 78-79 Content of carbohydrates seem to be quite similar for all extracts. Is this because of high levels of small sugars that were not precipitable with alcohol. What other than proteins, carbohydrates, and phenolic compounds are in the extracts?
Ln 240-243 ‘This result demonstrates that toxic compounds present in the aqueous extract were separated from the non-toxic ones after the ethanol precipitation, remaining in the S-EA fraction, while P-EA accumulated only polysaccharides that usually do not cause harm to cells.’
Can this statement be justified, given that cytotoxicity against normal and tumour cells was not tested over the same timescales? Polyphenol content of P-EA was like EA so could P-EA at high concentrations have been detrimental to normal cells over 72 hours?
Ln 245 Why different timescales for MCF7 tumoral cells and HepG2 cells?
Ln 316-320 The findings show that P-EA can act against cancer cells in vitro as do many plant extracts. You need to discuss how this could be of use in vivo or speculate about possible modes of action.
Ln 381 How many individual extracts of the fruits of Liquidambar styraciflua L. were prepared. How reproducible were the extracts in terms of yield and composition?
Ln 382-383 What were conditions for drying of the fruits? Give more details of extraction procedures used or cite appropriate references.
Author Response
Dear editor and reviewers,
We thank for the revision of this manuscript and for the suggestions that help us to improve the quality of this manuscript. All the requested observations were answered, and the responses are marked in blue. The corrections made to the manuscript are marked in red.
Reviewer 1
Comments and Suggestions for Authors
The authors have conducted studies to investigate the chemical composition and biological potential of extracts obtained from the fruits of Liquidambar styraciflua L. An aqueous extract (EA) was prepared and mixed with ethanol to give precipitable (P-EA) and soluble (S-EA) fractions which comprised of proteins, phenolic compounds, and carbohydrates in different proportions. P-EA contained pectic polysaccharides, such as acetylated and methyl esterified homogalacturonans together with arabinogalactan while S-EA contained phenolic acids and terpenes such as gallic acid, protocathecuic acid, liquidambaric acid, combretastatin and atractyloside A. EA, P-EA,and S-EA had antioxidant activity. They report that S-EA was toxic to normal and tumour cells whereas P-EA had low toxicity for normal cells in culture but high toxicity against tumour cells and was therefore promising candidate for studies against cancer.
However, the comparisons between the effects of EA, P-EA, S-EA on cells are not alike. Studies with HepG2 cells were done over 24 and 48 hours whereas the study with MCF7 tumoral cells were done at 24, 48 and 72 hours with IC50 measured at 72 hours. What is to say that similar toxicities would not have been evident with HepG2 cells after 72 hours?
Response: In this study we used three different cell lines: non-tumorigenic HepG2 (to evaluate hepatotoxicity), breast tumoral MCF7 cells (to evaluate antitumoral effect on a breast cancer model), and the normal breast cell MCF10A (as control).
Cytotoxicity tests are normally performed at 24h and 48h. Therefore, our preliminary tests were performed incubating the extracts (EA, P-EA, and S-EA) with the three cell lines for this two time-points. As we observed that P-EA presented no toxicity at all to the HepG2 cells (cell viability showed no reduction), but presented slight toxicity to MCF7 tumoral cells, we decided to increase the treatment to 72h only for the breast tumoral cells and use the breast normal cells as control, considering that tumoral and normal breast cells are similar.
And we agree with the reviewer that the other two extracts (EA and S-EA) that presented toxicity to HepG2 cells at 24h and 48h would probably cause toxicity to such cells at 72h, however we did not see necessity to prove this point, because in this case we used the normal MCF10A cells to compare with MCF7 after 72h. And we had already observed that the EA and S-EA extracts are too toxic to any cell line and therefore they should not be further investigated as antitumoral drug.
IC50 was calculated only for the graphs that presented a curve fit that converged into dose-response.
Also, the authors need to have tested against a range of tumour cells rather than only one to make a robust statement that the extracts had anti-tumour actions. With one cell type the findings could be by chance.
Response: We agree with the reviewer that if more cell lines were studied, the result could be more robust as evidence of potential anticancer drug. However, there are some points that should be considered because we believe that the investigation performed with only one tumoral cell line do not invalidate our results.
- To test many different cell lines, a laboratory should have structure to manage all the cell lines, different culture medium, staff for helping with the cell culture maintenance, and financial support to manage all the costs required to keep this lab structure.
- Each tumoral cell line may have different characteristics and therefore, a drug/extract may be toxic to one tumoral cell line and do not present toxicity to other cell line, which do not impede a drug/extract to be considered as potential anticancer investigation.
- The present study showed the extraction of different compounds, the chemical characterization of secondary metabolites and polysaccharides, which requires sophisticated equipment such as LC-MS, GC-MS, NMR, and HPSEC. The chemical investigation of the compounds of L. styraciflua extracts comprises a plenty of experiments that is unusual for plant extract studies, and therefore this study contributes with important information to the scientific community. The biological investigation of this study is indeed a preliminary result, however the amount of information about the chemical characteristics of the extracts summed to the effect on the three cell lines gives a direction for further studies. Therefore, we did not see necessity to evaluate the extracts in many different tumoral cells for this study.
Proteins, carbohydrates, and phenolic compounds comprise at best fifty percent of the material in EA, P-EA, S-EA extracts. What is the rest?
Response: Nutrition analysis of different plants usually also find vitamins, lipids and fatty acids, minerals, fibers. However the intend of this study was not to determine all the nutritional components of the extracts, but to investigation the most biological active such as the secondary metabolites and polysaccharides.
What information presented is novel? What is different about use of the fruit instead of bark and stem as a source of bioactive compounds? Much of presented data seems to be confirmatory of what is found in bark and stem. Are they more readily available from fruit, are the yields better, or do the extracts have unexpected properties about effects on cells.
Response: As the literature have already presented information about the bark and stem, we decided to investigate the fruits, which are available, and their use do not cause harm to the trees. The use of fruits is environmentally correct. Furthermore, the yield of aqueous extract obtained from leaves (~7.3%) and fruits (~7.6%) is the same, and the yield of fruits is almost 7 times higher than the yield of stems (~1.7%), which justifies the use of fruits.
As it stands EA, P-EA, S-EA extracts each are mixtures of ill-defined factors that can influence the metabolism of tumour cells in culture, and P-EA is touted as having most potential. However, there are hundreds if not thousands of plant extracts which show similar properties. So, what makes this so different. The authors need to show that P-EA can be effective in a cancer model in vivo or show mechanisms by which P-EA can block growth of tumour cells in vitro which might indicate potential routes of action in vivo.
Response: Indeed, there are a plenty of other plants with the same properties, however this plant is traditionally used and consumed in Chinese medicine (as it was described in lines 36-37), which justifies its study and further investigation about its toxicity and properties. Natural products always served as vital resources for cancer therapy (e.g., Vinca alkaloids, camptothecin, paclitaxel, etc.). Furthermore, the success of phytotherapy is due to phytochemicals and herbal mixtures act multi-specifically, i.e., they attack multiple targets at the same time. Therefore, we do find important that popularly used plants and their extracts should be investigated to confirm their benefits and to investigate their toxicity as it was demonstrated in this study.
The discussion of our results was improved, and some possible mechanisms of action were described. The new text was added to lines 262-266; 272-281; and 310-316, and the in vivo studies to profound our preliminary conclusions could be performed in future investigations.
Ln 78-79 Content of carbohydrates seem to be quite similar for all extracts. Is this because of high levels of small sugars that were not precipitable with alcohol. What other than proteins, carbohydrates, and phenolic compounds are in the extracts?
Response: Yes, the carbohydrate content of P-EA and S-EA, besides with similar amounts, comprises different molecules. P-EA comprises high molecular weight carbohydrates such as pectic polysaccharides, while S-EA contains oligosaccharides, sucrose, fructose and glycosides.
As we mentioned in the previous answer: besides carbohydrates, proteins and phenolic compounds, plants usually contain vitamins, lipids and fatty acids, minerals, and fibers.
Ln 240-243 ‘This result demonstrates that toxic compounds present in the aqueous extract were separated from the non-toxic ones after the ethanol precipitation, remaining in the S-EA fraction, while P-EA accumulated only polysaccharides that usually do not cause harm to cells.’
Can this statement be justified, given that cytotoxicity against normal and tumour cells was not tested over the same timescales?
Response: No, because we tested the extracts on tumoral (MCF7) and normal (MCF10A) cells for the same timescale (72h).
Polyphenol content of P-EA was like EA so could P-EA at high concentrations have been detrimental to normal cells over 72 hours?
Response: This test was not performed; however, we compared the same concentrations of both extracts on HepG2 cells for 24h and 48h and observed not toxicity at all to the cells treated with P-EA. Even when the normal breast cells (MCF10A) were incubated with both extracts for 24h, 48h, and 72h, the toxicity was not observed. Therefore, we believe that toxic compounds were removed from P-EA extract and its potential to reduce tumoral cell viability may be related to other mechanism of action, different from the toxic property of EA and S-EA extracts.
Ln 245 Why different timescales for MCF7 tumoral cells and HepG2 cells?
Response: Cytotoxicity tests are normally performed at 24h and 48h. Therefore, our preliminary tests were performed incubating the extracts (EA, P-EA, and S-EA) with the three cell lines for this two time-points. As we observed that P-EA presented no toxicity at all to the HepG2 cells (cell viability showed no reduction), but presented slight toxicity to MCF7 tumoral cells, we decided to increase the treatment to 72h only for the breast tumoral cells and use the breast normal cells as control, considering that tumoral and normal breast cells are similar.
Ln 316-320 The findings show that P-EA can act against cancer cells in vitro as do many plant extracts. You need to discuss how this could be of use in vivo or speculate about possible modes of action.
Response: The discussion of our results was improved, and some possible mechanisms of action were described. The new text was added to lines 262-266; 272-281; and 310-316.
Ln 381 How many individual extracts of the fruits of Liquidambar styraciflua L. were prepared. How reproducible were the extracts in terms of yield and composition?
Response: For this study, the intention was to identify the chemical compounds present in the extracts and their effect on tumoral and normal cells. Comparison of yields of different extractions was not performed because other variants should be considered such as: fruit origin and fruit harvest season.
Ln 382-383 What were conditions for drying of the fruits? Give more details of extraction procedures used or cite appropriate references.
Response: The fruits were dried at 25oC, for 14 days. More details were added to section “3.2 Extraction Procedures”.

Reviewer 2 Report
Dear authors,
I was asked to provide a peer-review report of the manuscript “Chemical evaluation of Liquidambar styraciflua L. fruits extract and their potential as anticancer drugs” by Pozzobon et al.
The manuscript aims to provide new insights into the chemical content of three extracts from a traditional Chinese medicinal plant and evaluates their potential to influence cell growth of two different tumor cell lines as well as one normal breast epithelial cell line.
First of all, the aim and the methodology is well suited for the journal “Molecules”. I have some minor comments on the manuscript but appreciate the methodology and thoroughness of this study.
I have one general comment before I start to go into detailed evaluation: The manuscript lacks a “discussion” section. I guess it was a mistake and the authors wanted to combine “results and discussion”. A manuscript which is not discussing the results is not publishable.
I will give my further comments the following part.
1. In line 30 there is the plant family name in all capital letters. In my opinion this is not necessary by use of a plant family name within a sentence and I would therefore recommend to capitalize only the first letter.
2. What is meant by g%? Do you mean % (g/g) or % (w/w)? Please clarify and correct throughout the text.
3. How can the three extracts contain “similar amounts” (line 78) but “counted the greatest part of EA and P-EA”? Is this not contradictory or did you refer absolute mass and relative amount, meaning that the P-EA extract is much lower in total weight than the S-EA extract. This would also be not understandable in the light of line 69 (“similar yield values”). Please clarify!
4. I like your conclusions in the paragraph from lines 120 – 132!
5. You can be even more specific in line 130 and name rhamnogalacturonan-I when referring to GalA and Rha.
6. I think your statement in lines 211 - 213 can be more specific. There are a lot of known antioxidants, why should you add another to the list?
7. I might have missed it, but do you have an explanation for the increase in toxicity at 1000 µg/ml (P-EA; lines 254 -256)? Afterwards you wrote about low toxicity of polysaccharides. Please clarify!
8. Your main identified components in the polysaccharide extract are pectins and arabinogalactans. Both groups are also known as interaction partners to human galectins, which are on the surface of some cancer cell lines. For example, Schöll-Naderer et al. (2020) https://doi.org/10.1016/j.carres.2019.107903 showed that Panc-1 cells, which you also discussed, are inhibited in some metastasis-related processes by the interaction between AG and galectin-3. There are other works like for example, Pfeifer et al. (2021) https://doi.org/10.3390%2Fijms22084058 who showed binding for AGs and different galectins. In my opinion, this would add to your discussion another dimension that not only the growth of cancer cell lines is inhibited but also metastasis-related processes can be reduced.
9. I wondered why you used tap water for dialysis (in line 387) and not demineralized water. Can you explain this clearer? It would also be nice to read how often you exchanged the water and how long the dialysis was in total.
In conclusion, the manuscript is well-written and I recommend publication in “Molecules” after minor revision.
All the best!
Author Response
Dear editor and reviewers,
We thank for the revision of this manuscript and for the suggestions that help us to improve the quality of this manuscript. All the requested observations were answered, and the responses are marked in blue. The corrections made to the manuscript are marked in red.
Reviewer 2
Comments and Suggestions for Authors
Dear authors,
I was asked to provide a peer-review report of the manuscript “Chemical evaluation of Liquidambar styraciflua L. fruits extract and their potential as anticancer drugs” by Pozzobon et al.
The manuscript aims to provide new insights into the chemical content of three extracts from a traditional Chinese medicinal plant and evaluates their potential to influence cell growth of two different tumor cell lines as well as one normal breast epithelial cell line.
First of all, the aim and the methodology is well suited for the journal “Molecules”. I have some minor comments on the manuscript but appreciate the methodology and thoroughness of this study.
Response: Thank you very much!
I have one general comment before I start to go into detailed evaluation: The manuscript lacks a “discussion” section. I guess it was a mistake and the authors wanted to combine “results and discussion”. A manuscript which is not discussing the results is not publishable.
Response: Indeed, it was a mistake, we forgot to describe that section 2 is in fact the combined “Results and Discussion”. This was corrected in the text. We believe that is more interesting to report a result and then add its correspondent discussion at the same part of the text to avoid repetition.
I will give my further comments the following part.
- In line 30 there is the plant family name in all capital letters. In my opinion this is not necessary by use of a plant family name within a sentence and I would therefore recommend to capitalize only the first letter.
Response: Ok, we agree and changed the text as requested.
- What is meant by g%? Do you mean % (g/g) or % (w/w)? Please clarify and correct throughout the text.
Response: Actually, it was a typo, the correct is % (g of extract/100 g of fruits). This was corrected and clarified in the text.
- How can the three extracts contain “similar amounts” (line 78) but “counted the greatest part of EA and P-EA”? Is this not contradictory or did you refer absolute mass and relative amount, meaning that the P-EA extract is much lower in total weight than the S-EA extract. This would also be not understandable in the light of line 69 (“similar yield values”). Please clarify!
Response: Indeed, the text was not clear. It was re-written. However, the fractions P-EA and S-EA, that were derived from EA extract do contain similar yields: P-EA (6.14 g) and S-EA (6.64 g), totalizing ~3% of fraction in 100 g of fruits.
- I like your conclusions in the paragraph from lines 120 – 132!
Response: Thank you.
- You can be even more specific in line 130 and name rhamnogalacturonan-I when referring to GalA and Rha.
Response: Yes, we agree, however we prefer to not specify at this part of the text, based only on the monosaccharide composition. Therefore we added the specific name “rhamnogalacturonan-I” in line 156, after the confirmation of NMR signals.
- I think your statement in lines 211 - 213 can be more specific. There are a lot of known antioxidants, why should you add another to the list?
Response: Indeed, we agree there are a lot of known antioxidants, there is no sense in search for new antioxidant compounds. The search is to find plants with antioxidant properties (containing known antioxidant compounds) that could also present another therapeutic property. The sentence was modified, and it is more specific in lines 214-215.
- I might have missed it, but do you have an explanation for the increase in toxicity at 1000 µg/ml (P-EA; lines 254 -256)? Afterwards you wrote about low toxicity of polysaccharides. Please clarify!
Response: Yes, the text was re-written to clarify this doubt (please check lines 256-266). The fraction P-EA (pectic polysaccharides) can not be considered toxic to MCF7 cells (tumoral cell line) after 48h of incubation at 1000 µg/mL, according to ISO 10993-5:2009, because it reduced less than 30% (~25%) of cell viability.
This reduction is considered as small and/or no toxicity and comparing the results of P-EA extract with S-EA extract, we conclude that S-EA extract is indeed farther more toxic than P-EA extract, at 48 h of incubation.
Reference: ISO 10993-5 (2009) Biological Evaluation of Medical Devices. Part 5: Tests for in Vitro Cytotoxicity.
- Your main identified components in the polysaccharide extract are pectins and arabinogalactans. Both groups are also known as interaction partners to human galectins, which are on the surface of some cancer cell lines. For example, Schöll-Naderer et al. (2020) https://doi.org/10.1016/j.carres.2019.107903 showed that Panc-1 cells, which you also discussed, are inhibited in some metastasis-related processes by the interaction between AG and galectin-3. There are other works like for example, Pfeifer et al. (2021) https://doi.org/10.3390%2Fijms22084058who showed binding for AGs and different galectins. In my opinion, this would add to your discussion another dimension that not only the growth of cancer cell lines is inhibited but also metastasis-related processes can be reduced.
Response: Thank you for the suggestion! Indeed, the two references and others related to galectins improved our discussion. However, since a metastasis was not experimentally tested in this study, we prefer to keep the discussion to the reduction on cell viability. The new text was added to lines 262-266; 272-281; and 310-316.
- I wondered why you used tap water for dialysis (in line 387) and not demineralized water. Can you explain this clearer? It would also be nice to read how often you exchanged the water and how long the dialysis was in total.
Response: Dialysis was performed for 24h as it was explained in this section 3.2 (Extraction procedures). In fact, tap water was used because the dialysis procedure was not performed in a closed system. It was done in an open system, using tap water, for 24h. This procedure guarantee to remove small Mw compounds (especially oligosaccharides and ethanol residue) from the polysaccharide fraction, however some salt content may be present. As soon as the salt content do not interfere in our experiments, this was not a problem. This information was clarified in line 410.
In conclusion, the manuscript is well-written and I recommend publication in “Molecules” after minor revision.
All the best!

Reviewer 3 Report
Manuscript Title: Chemical evaluation of Liquidambar styraciflua L. fruits ex- 2 tracts and their potential as anticancer drugs
Manuscript Number: molecules-2104438
Article Type: Article
Comments:
The manuscript “Chemical evaluation of Liquidambar styraciflua L. fruits ex- 2 tracts and their potential as anticancer drugs” by Fernanda Ribeiro Smiderle and co-workers designed to investigate the chemical composition and biological potential of extracts obtained from the fruits of this plant. These structures were analysed by using spectroscopic analysis. The fraction S-EA is very cytotoxic to the cancer and normal cell lines. Interestingly, the results suggest that the fraction P-EA is a promising candidate for further studies against series of cancers since high cytotoxicity to tumor cells and low toxicity to normal cells.
I thoroughly read the entire manuscript from beginning to till the end. I have seen some of the corrections in this manuscript as small grammatical mistakes, typos, and spacing between the words. And all the corrections are highlighted in the attached draft. The author needs to be revised all the corrections in the attached draft. After careful reading, I am considering this manuscript needs a minor revision. Please re-submit the manuscript with the improved and revised version.
Modest items that need to be addressed include:
1. Manuscript English grammar needs to be improved and corrected.
2. In the abstract ‘chromatographic, spectroscopy and spectrometry methods’, this sentence need be changed.
3. The results part in page 2, yield of 7.60 g%, 7.6 g is a quantitative yield need to calculate % of yield. Please correct accordingly in the whole manuscript.
4. The yellow color highlights should be corrected in the attached draft.
5. It would be good if you add full spectra of HSQC-DEPT spectra.
6. References typos highlighted in the attached draft and need to be corrected

Author Response
Dear editor and reviewers,
We thank for the revision of this manuscript and for the suggestions that help us to improve the quality of this manuscript. All the requested observations were answered, and the responses are marked in blue. The corrections made to the manuscript are marked in red.
Reviewer 3
Comments and Suggestions for Authors
Manuscript Title: Chemical evaluation of Liquidambar styraciflua L. fruits ex- 2 tracts and their potential as anticancer drugs
Manuscript Number: molecules-2104438
Article Type: Article
Comments:
The manuscript “Chemical evaluation of Liquidambar styraciflua L. fruits ex- 2 tracts and their potential as anticancer drugs” by Fernanda Ribeiro Smiderle and co-workers designed to investigate the chemical composition and biological potential of extracts obtained from the fruits of this plant. These structures were analysed by using spectroscopic analysis. The fraction S-EA is very cytotoxic to the cancer and normal cell lines. Interestingly, the results suggest that the fraction P-EA is a promising candidate for further studies against series of cancers since high cytotoxicity to tumor cells and low toxicity to normal cells.
I thoroughly read the entire manuscript from beginning to till the end. I have seen some of the corrections in this manuscript as small grammatical mistakes, typos, and spacing between the words. And all the corrections are highlighted in the attached draft. The author needs to be revised all the corrections in the attached draft. After careful reading, I am considering this manuscript needs a minor revision. Please re-submit the manuscript with the improved and revised version.
Modest items that need to be addressed include:
- Manuscript English grammar needs to be improved and corrected.
Response: The grammar was revised and improved as requested.
- In the abstract ‘chromatographic,spectroscopy and spectrometry methods’, this sentence need be changed.
Response: Ok, this sentence was re-written: “For the chemical evaluation, it was used mainly liquid and gas chromatography, plus NMR, and colorimetric methods.”
- The results part in page 2, yield of 7.60 g%, 7.6 g is a quantitative yield need to calculate % of yield. Please correct accordingly in the whole manuscript.
Response: Actually, it was a typo, the correct is % (g of extract/100 g of fruits). This was corrected and clarified in the text.
- The yellow color highlights should be corrected in the attached draft.
Response: Ok, thank you for your careful revision. The typos were corrected, and all the manuscript, including the references were revised and corrected.
- It would be good if you add full spectra of HSQC-DEPT spectra.
Response: The full spectrum of sample P-EA was added to the supplementary material, in different scales (Fig. S1), to show that polysaccharide signals were present and there was no contamination of other compounds. We believe that the resolution of the full spectrum was poor due to the presence of too many empty spaces, therefore we decided to keep the partial spectrum on the manuscript to evidence the important signals.
- References typos highlighted in the attached draft and need to be corrected
Response: Ok, thank you for your careful revision. The typos were corrected, and all the manuscript, including the references were revised and corrected.

Round 2
Reviewer 1 Report
The issues raised have been addressed and dealt with in a satisfactory manner.